# RETRACTED: Empowering Low- and Middle-Income Countries to Combat AMR by Minimal Use of Antibiotics: A Way Forward

**DOI:** 10.3390/antibiotics12101504

**Published:** 2023-10-02

**Authors:** Mohammed Kanan, Maali Ramadan, Hanan Haif, Bashayr Abdullah, Jawaher Mubarak, Waad Ahmad, Shahad Mari, Samaher Hassan, Rawan Eid, Mohammed Hasan, Mohammed Qahl, Atheer Assiri, Munirah Sultan, Faisal Alrumaih, Areej Alenzi

**Affiliations:** 1Department of Clinical Pharmacy, King Fahad Medical City, Riyadh 12211, Saudi Arabia; 2Department of Pharmacy, Maternity and Children Hospital in Rafha, Rafha 76312, Saudi Arabia; maraalshammari@moh.gov.sa (M.R.); halshammari19@moh.gov.sa (H.H.); balznezi@moh.gov.sa (B.A.); pharm.d.jawaher@gmail.com (J.M.); 3Department of Pharmacy, King Khalid University, Abha 61421, Saudi Arabia; waadalqaedi_0197@hotmail.com (W.A.); shahd14121@gmail.com (S.M.); 4Department of Clinical Pharmacy, Jazan College of Pharmacy, Jazan 82726, Saudi Arabia; 201600591@stu.jazanu.edu.sa; 5Department of Pharmacy, Nahdi Company, Tabuk 47311, Saudi Arabia; alanazi.re@nahdi.sa; 6Department of Pharmacy, Armed Forces Hospital Southern Region, Mushait 62562, Saudi Arabia; mhm1417@hotmail.com (M.H.); assiriatheer19@gmail.com (A.A.); 7Department of Pharmacy, Najran Armed Forces Hospital, Najran 66256, Saudi Arabia; phy.eissa@nafh.med.sa; 8Al-Adwani General Hospital, Taif 26523, Saudi Arabia; pharm.muneera77@hotmail.com; 9Department of Pharmacy, Northern Border University, Rafha 76313, Saudi Arabia; faisal.alrumaih@hotmail.com; 10Department of Infection Control and Public Health, Regional Laboratory in Northern Border Region, Arar 73211, Saudi Arabia; araalenzi@moh.gov.sa

**Keywords:** AMR, antibiotics, low-middle-income countries, empowering, combat

## Abstract

Antibiotic overuse poses a critical global health concern, especially in low- and middle-income countries (LMICs) where access to quality healthcare and effective regulatory frameworks often fall short. This issue necessitates a thorough examination of the factors contributing to antibiotic overuse in LMICs, including weak healthcare infrastructure, limited access to quality services, and deficiencies in diagnostic capabilities. To address these challenges, regulatory frameworks should be implemented to restrict non-prescription sales, and accessible point-of-care diagnostic tools must be emphasized. Furthermore, the establishment of effective stewardship programs, the expanded use of vaccines, and the promotion of health systems, hygiene, and sanitation are all crucial components in combating antibiotic overuse. A comprehensive approach that involves collaboration among healthcare professionals, policymakers, researchers, and educators is essential for success. Improving healthcare infrastructure, enhancing access to quality services, and strengthening diagnostic capabilities are paramount. Equally important are education and awareness initiatives to promote responsible antibiotic use, the implementation of regulatory measures, the wider utilization of vaccines, and international cooperation to tackle the challenges of antibiotic overuse in LMICs.

## 1. Introduction

Antibiotic overuse entails the inappropriate and excessive utilization of antibiotics, encompassing unwarranted prescriptions and administration, as well as the mishandling and excessive consumption of these vital medications through self-medication or leftover usage [1,2]. The overuse of antibiotics has emerged as a pivotal global health concern, bearing extensive ramifications and significantly affecting individuals, healthcare systems, and overall public health. Particularly pronounced in low- and middle-income countries (LMICs), the issue of antibiotic overuse is exacerbated by factors such as deficient healthcare infrastructure, inadequate regulatory frameworks, restricted access to quality medical services, limited diagnostic capacities, and insufficient awareness and education concerning proper antibiotic utilization [3,4,5,6]. As a result, the improper utilization of antibiotics is becoming widespread, significantly contributing to the escalation of antimicrobial resistance (AMR) [7,8]. AMR stands as a worldwide crisis, endangering the efficacy of antibiotics and presenting a substantial menace to public health [9,10]. The excessive or incorrect use of antibiotics can lead to the development of bacterial mechanisms that elude the drugs’ impact, ultimately leading to their diminished effectiveness [11]. This phenomenon results in the failure of treatments for individual patients and obstructs the effective management of infectious diseases on a broader population scale. In LMICs, where healthcare resources are already under pressure, the emergence and propagation of antibiotic-resistant pathogens worsen the difficulties encountered by healthcare providers and render healthcare costs unmanageable [12]. Addressing the global issue of antibiotic overuse requires a comprehensive approach involving various measures. These include improving healthcare infrastructure, enforcing effective regulatory frameworks, ensuring access to quality healthcare, enhancing diagnostic capabilities, and expanding preventive interventions. Educating healthcare providers, patients, and the public about appropriate antibiotic use is crucial. Interventions should focus on responsible prescribing practices, implementing antimicrobial stewardship programs, and changing behaviors related to antibiotic consumption.

This study aims to explore the complexities of antibiotic overuse in LMICs, highlighting their unique challenges and proposing strategies to tackle the problem. By advocating for global collaboration, the article seeks to combat antibiotic overuse and protect the effectiveness of antibiotics for future generations.

## 2. Trends of Consumption in LMICs

Over the last few decades, antibiotic consumption patterns in LMICs have displayed a notable upward trend. A multinational survey that analyzed antibiotic consumption across 76 countries from 2000 to 2015 exposed a significant global increase of 65%. This surge was primarily fueled by a remarkable upswing in LMICs, which experienced a striking rise of 114% [1]. Furthermore, a systematic analysis of national surveys conducted across 73 countries underscored a consistent and upward trajectory in antibiotic usage among children aged under five years old. This trend was observed between 2005 and 2017 and was particularly pronounced in LMICs [13].

The escalating use of antibiotics in LMICs underscores a troubling pattern of their misapplication. A noteworthy concern is the substantial uptick in the consumption of antibiotics labeled as “Watch” in the WHO’s framework. These antibiotics carry a higher risk of resistance development and include fluoroquinolones, macrolides, carbapenems, and glycopeptides. Worryingly, these crucial antibiotics are frequently prescribed and obtained without proper prescriptions in LMICs. Global data spanning from 2000 to 2015 expose a staggering 90.9% surge in the consumption of “Watch” antibiotics, with a far more disconcerting 164% surge witnessed in LMICs. WHO advocates that a minimum of 60% of all antibiotics used belong to the “Access” group, comprising affordable, narrow-spectrum agents. Regrettably, an increasing number of countries are failing to meet this target. In 2015, out of 76 countries, 42 displayed lower proportions of “Access” antibiotics [2]. These consumption patterns highlight a substantial problem of improper antibiotic usage in LMICs [3,5,6,7]. Frequently, antibiotics are prescribed or obtained without prescriptions for conditions that either resolve on their own or are caused by viruses, making antibiotic treatment unnecessary. This encompasses ailments such as respiratory tract infections like common colds or bronchitis, as well as diarrheal diseases, particularly among children [14]. An examination of data from simulated patient studies carried out in India, China, and Kenya, with a focus on conditions where antibiotics are not recommended (such as acute diarrhea, respiratory tract infection, pulmonary tuberculosis, angina, and asthma), unveiled that approximately 30–50% of patient–provider interactions led to the prescription or direct distribution of one or more antibiotics [8,9,10]. A concerning fact emerges from India, where a staggering 47.6% (with a 95% confidence interval of 26.8–54.0) of all antibiotics prescribed or dispensed in 2392 visits were categorized within the “Watch” group. This highlights the widespread utilization of antibiotics with a higher potential for resistance [12,13,14,15]. Moreover, in LMICs there is a notable utilization of discouraged fixed-dose combinations of antibiotics. This is often carried out without sufficient evidence of their effectiveness and without considering the potential risks linked to incorrect dosing [16,17]. These non-recommended combinations frequently include antibiotics classified as “Watch,” such as macrolides or quinolones. This further worsens the issue of antibiotic overuse by introducing these potent antibiotics in a less regulated and controlled manner [17].

The COVID-19 pandemic has exacerbated the problem of excessive antibiotic usage, increasing worries about the swift emergence and spread of bacterial strains that are resistant to antibiotics [18]. In India, a lower-middle-income country and one of the world’s leading consumers of antibiotics, one study employed an interrupted time series analysis of pharmaceutical sales data from the private market spanning 2018 to 2020 [19]. It estimated that the first wave of the epidemic led to additional sales of non-pediatric antibiotic formulations totaling 216.4 million doses. A notable focus was on the surplus sales of azithromycin, which accounted for 38.0 million doses, equivalent to more than 6 million treatment courses [20]. This surge in sales likely reflects the inappropriate repurposing of azithromycin for the treatment of COVID-19 cases. Similar patterns have also emerged in other LMICs [21,22]. Moreover, an examination of pharmaceutical sales data from 71 countries between 2020 and 2022 revealed a noticeable correlation between the sales of penicillin, cephalosporins, and macrolides, and the reported COVID-19 cases across various continents [23,24].

## 3. Factors Associated with Extensive Use

In LMICs, the overuse of antibiotics is propelled by an intricate interplay of factors that collectively contribute to elevated rates of antibiotic consumption (Figure 1).

Comprehending these underlying drivers is imperative for formulating effective strategies to tackle this issue. Many LMICs grapple with limited access to healthcare and facilities that are understaffed and lack adequate resources [25,26,27]. Consequently, patients frequently resort to seeking care from informal healthcare providers or obtaining antibiotics directly from pharmacies without the need for a prescription [26]. The absence of qualified healthcare providers and proper diagnostic tools can contribute to antibiotic overuse, as precautionary measures or inaccurate diagnoses might prompt unnecessary antibiotic prescriptions [28,29,30]. Cultural beliefs and societal expectations also play a role, as patients may pressure healthcare providers to prescribe antibiotics, mistakenly believing they are a cure-all for various illnesses [31,32,33]. Furthermore, cultural norms such as self-medication and the practice of sharing antibiotics within families also play a role in contributing to antibiotic overuse [34,35,36,37,38]. A lack of public awareness regarding proper antibiotic use, the implications of antibiotic resistance, and the significance of completing a full treatment course is a prevailing issue in numerous settings, including LMICs [39,40]. Misconceptions about antibiotics perpetuate their overuse, with many people considering them a quick fix for a range of ailments. The fragmented nature of healthcare systems in LMICs presents challenges in disseminating and implementing standardized guidelines [41,42]. The absence of coordination among various healthcare providers, encompassing both public and private sectors, leads to inconsistent prescribing practices and restricted adherence to recommended guidelines [37,39,43]. Moreover, inadequate regulatory frameworks and financial incentives originating from profit-driven pharmaceutical companies also contribute to the issue of antibiotic overuse [44]. The convenient accessibility of antibiotics without a prescription from community pharmacists and various types of medicine vendors intensifies the problem even more [25]. To effectively tackle these challenges, a multifaceted approach is essential. In the subsequent sections of this article, we will delve into significant obstacles and explore potential solutions. Moreover, we will offer concrete examples of successful strategies implemented across different contexts.

## 4. Issues and Mitigations

### 4.1. Regulatory Framework

In LMICs, antibiotics are easily obtainable without requiring a prescription, primarily due to the absence of stringent regulations or the insufficient enforcement of existing laws [26,45]. Nevertheless, the mere introduction of regulations designed to limit antibiotic sales without prescriptions does not always lead to a decrease in antibiotic consumption [46]. Chile, despite being a high-income country (HIC), presents a successful model for the implementation of antibiotic use regulations that could be adapted for LMICs. In Chile, strict enforcement of antibiotic use regulations, accompanied by a public awareness campaign and pharmacy support, led to a significant reduction in antibiotic sales. In contrast, in Venezuela, regulations introduced without proper enforcement and awareness efforts had no impact on sales. Some LMICs have restricted non-prescription antibiotic sales, resulting in decreased sales of those antibiotics, but often leading to increased sales of unrestricted alternatives like penicillin [46,47]. Likewise, in 2018, India implemented a ban on specific antimicrobial fixed-dose combinations (FDCs), leading to a notable reduction in the sales of these prohibited FDCs [15]. Nevertheless, the sales of combinations containing components from the same drug classes as those found in the banned FDCs actually increased. Additionally, the sales of other non-banned formulations of these combinations also experienced a significant rise, effectively counteracting the impact of the ban. In a recent systematic review, the effects of regulations aiming to curb over-the-counter antibiotic sales in LMICs were examined [47,48]. The review analyzed 15 studies from 10 LMICs, revealing that countries combining law enforcement with awareness campaigns and stakeholder engagement achieved more lasting success in curbing over-the-counter antibiotic sales. Informal healthcare workers, without formal training, are crucial in delivering healthcare in many LMICs, particularly in rural and underserved areas [48,49]. While highly sought after by clients, these workers lack formal recognition in regulatory frameworks. Informal healthcare workers make up 55% of providers in India and account for 75% of primary care visits, despite not being officially acknowledged [49,50]. In Bangladesh, around 96% of rural healthcare providers are classified as informal workers [51]. Multiple studies conducted in LMICs have demonstrated that informal healthcare workers often prescribe broad-spectrum antibiotics [52,53]. Trained medical professionals often link antibiotic overuse to informal healthcare workers’ practices [54]. In many LMICs, informal healthcare workers play a vital role due to fragmented health systems. Strategies to improve antibiotic use should involve integrating these workers through recognition, training, and regulation, as demonstrated by an Indian trial [49]. Training did not significantly alter antibiotic use. Regulations on non-prescription sales in LMICs effectively cut use, particularly with strong enforcement. Comprehensive interventions involving all stakeholders also reduced consumption. Enforcement may limit rural access; contextual solutions are needed. Moreover, appropriate policies and proper implementation may help in this regard. Different laws for urban/rural areas could help [44]. Furthermore, it is valuable to contemplate integrating informal healthcare workers into the legal framework in rural regions. Providing them with suitable training in antimicrobial prescription could enable them to prescribe and dispense “Access” antibiotics and basic “Watch” antibiotics. This approach seeks to empower informal healthcare workers while promoting prudent antibiotic usage in such contexts.

### 4.2. Accessible Diagnostic Facilities

Diagnostic uncertainty stands as a significant factor driving antibiotic overuse in primary care, impacting both HICs and LMICs [55]. Point-of-care (POC) diagnostic tests hold promise in aiding infection diagnosis during clinical encounters, assisting healthcare providers in informed antibiotic use decisions. These tests also enhance patient communication by enabling evidence-based prescriptions. In LMICs, implementing POC tests for malaria has resulted in decreased overtreatment with anti-malarial drugs [56,57]. Yet, it is important to acknowledge that antibiotic usage rose among patients who tested negative for malaria, given the lack of knowledge about alternative causes of fever [57]. This highlights the intricacies of decision making in syndromic infection management. Take the instance of patients in LMICs with symptoms of sexually transmitted infections: empirical treatment usually involves antibiotics for *Neisseria gonorrhoeae* and *Chlamydia trachomatis*, necessitating the use of two distinct antibiotics [58]. In LMICs, undifferentiated fever can result from bacterial (enteric fever, scrub typhus, leptospirosis) or viral (Dengue) infections [59]. POC tests ruling out single infections might not immediately alter antibiotic prescribing. Yet, in certain cases, they can decrease antibiotic usage. Tests detecting multiple pathogens could significantly cut overuse. Biomarker-based POC tests like C-reactive protein (CRP) have been explored in LMICs. A review assessed CRP’s effectiveness in lowering antibiotic use for febrile patients in LMICs [60]. CRP, when used with clinical symptoms and other POC tests, can cut antibiotic overuse for febrile patients. However, it is insufficient alone for bacterial/non-bacterial differentiation. Molecular POC tests are accurate but costly for LMICs [61]. Rapid antigen/antibody POC tests are cost-effective but less sensitive than molecular tests, and there is an urgent need for low-cost, high-sensitivity antigen/antibody lateral flow assays [62]. Recent advances in nanolabel detection show promise, matching molecular test accuracy [63]. Novel technologies can overcome POC test cost barriers in LMICs. Implementing affordable, accurate, and user-friendly POC tests for multiple pathogens could profoundly impact antibiotic overuse.

### 4.3. Scalable Stewardship Programs

Antimicrobial stewardship programs effectively cut antibiotic use in both hospital and non-hospital environments [64]. In HICs, despite the growth of stewardship programs in inpatient care settings [65] and where data that demonstrate the success of various interventions at local levels are still available [66,67], per capita antibiotic usage has shown only slight declines overall [64]. While there is some encouragement in per capita antibiotic consumption stabilizing in well-resourced regions, the reality persists that a significant portion of usage remains inappropriate or unnecessary [68,69,70]. This implies that effective national interventions should lead to reduce per capita antibiotic use. In LMICs, where antibiotic use is rising and AMR poses a greater mortality threat, the situation is concerning [71]; the AMR threat requires bold, adaptable interventions in LMICs. Centralized data collection, like Turkey’s system, can track trends, addressing limitations seen in highly resourced regions [72]. This centralized approach aids in achieving effective stewardship goals [73] and facilitates thorough assessments of wide-reaching interventions, like national policy shifts [74]. Both HICs and LMICs face staff shortages for stewardship. Non-physician providers can implement interventions and promote responsible antibiotic use. Pharmacists play a key role in implementing safeguards [75]. Pharmacist-led interventions have demonstrated substantial improvements in adherence to appropriate use guidelines in inpatient settings across various African nations [76,77,78]. Evidence supports training pharmacists for stewardship. With outpatient antibiotic use being prevalent, community pharmacists hold untapped potential, especially where prescriptions are not mandated. In LMICs, they are interested in community-based stewardship [79,80]. By engaging community pharmacists for guideline-aligned antibiotic dispensing, interventions can cover broad areas while being tailored to specific regions using existing community ties. Standardized education, for both patients and providers, relies on national standards and strong regulatory rules to prevent private interests from distorting guidelines. Conflicting resources create confusion among patients and providers. Robust regulations are essential to prevent drug manufacturers’ undue influence on education, a concern in some LMICs like Bangladesh and Nepal [81,82]. The most ambitious provider education could involve extra certification for antibiotic dispensing. Yet, due to political and practical hurdles, a more viable approach is widespread dissemination and clear messaging about existing national guidelines. Notably, mere guideline creation does not seem to significantly change practice patterns [68]. In LMICs, educational interventions are more effective when part of a comprehensive strategy [83]. For instance, a review in China found that education alone did not reduce antibiotic use, but when combined with feedback or regulatory measures, prescriptions decreased [84]. Robust IT infrastructure is essential for accessible guidelines, along with a plan for widespread dissemination through networks. Feedback mechanisms are vital for assessing guideline impact. In LMICs lacking national guidelines, the WHO AWaRe Antibiotic Book can be invaluable [85]. The WHO AWaRe Antibiotic Book provides accessible guidelines for common bacterial infections, with evidence citations and non-antibiotic management advice [86,87,88]. National guidelines establish care standards and frameworks for stewardship program implementation. In the US, CDC offers hospital stewardship program guidance that is replicable with minimal resources. Regulatory enforcement (e.g., joint commission requirement) is crucial [65] for universal uptake, underscoring the need for robust regulation in scalable interventions within healthcare systems.

### 4.4. Vaccination Drives

The threat of AMR and the rapid emergence of new pathogens highlights the importance of vaccines. With multidrug-resistant bacteria spreading and limited treatment options [89,90], vaccines can prevent illness and reduce antibiotic demand. This helps preserve antibiotics, offers cost savings, and enhances public health. Treating bacterial infections, especially resistant ones, can be costly due to complications and hospitalizations [91,92,93]. Vaccines can result in substantial cost savings by reducing hospitalizations and antibiotic use [94]. Expanding immunization programs, especially in regions with antibiotic overuse like LMICs, is vital in combating AMR. Vaccines have transformed our fight against infectious diseases, notably decreasing the burden of deadly conditions and addressing diseases that contribute to antibiotic misuse. *Streptococcus pneumoniae*, a major cause of pneumonia, is an illustrative example, causing significant global morbidity and mortality [95]. The introduction of polysaccharide-based and conjugate vaccines has notably lessened the impact of drug-susceptible and drug-resistant pneumococcal diseases, including invasive cases like meningitis and sepsis. These vaccines provide protection against a subset of the over 100 known serotypes that commonly cause disease. Although broader-coverage vaccines are desired to avoid the rise of non-vaccine serotypes becoming antibiotic-resistant, current pneumococcal vaccines have already reduced illness, subsequently decreasing antibiotic use [96]. A comprehensive study spanning 18 LMICs indicated that, with current coverage levels, pneumococcal conjugate vaccines (PCVs) prevent nearly 24 million episodes of antibiotic-treated illness annually in children under five years old [97]. Rotavirus and pneumococcal vaccinations could prevent millions of antibiotic-treated illnesses each year, particularly in cases of respiratory infections and diarrhea [7,14,98]. The extent to which vaccination indirectly reduces antibiotic usage is still uncertain and requires more investigation. A meta-analysis of 96 studies from 2019 suggests that current evidence lacks clear conclusions due to data limitations and methodological issues. However, recent analysis of a trial for an experimental respiratory syncytial virus vaccine showed promising results in reducing antibiotic usage [98,99]. Maternal vaccination could prevent around 3.6 antimicrobial prescription courses per 100 infants in HICs and 5.1 courses per 100 infants in LMICs [100,101,102]. This underlines the connection between vaccination and reduced antibiotic usage. An example is Pakistan, where vaccination curbed an outbreak of extensively drug-resistant typhoid cases in 2016 [103,104]. Since then, cases spread nationally and abroad, constituting a serious public health threat. XDR typhoid is caused by *Salmonella enterica* serovar Typhi, which is resistant to chloramphenicol, ampicillin, co-trimoxazole, fluoroquinolones, and third-generation cephalosporins, thus leaving very limited options for treating affected individuals(azithromycin, carbapenems, and tigecycline) [105,106]. The global misuse of oral azithromycin during the COVID-19 pandemic has endangered its efficacy. Moreover, carbapenem resistance is spreading in bacterial species like *Escherichia* and *Klebsiella* spp [107,108]. Pakistan is taking a proactive stance against XDR typhoid by initiating a campaign to vaccinate over 10 million children (9 months to 15 years) in high-risk areas. It is the first country to integrate the new typhoid conjugate vaccine (TCV) into its immunization program [109]. TCV is a single-dose injectable vaccine containing Vi polysaccharide conjugated to tetanus toxoid [110]. In a 2018–2019 cohort study in Sindh, TCV showed 55% effectiveness against suspected typhoid, 95% against culture-confirmed S. Typhi, and 97% against XDR S. Typhi [111]. A GAVI-supported mathematical modeling study indicates that implementing TCV for children up to 15 years in 73 countries could prevent 50 million drug-resistant typhoid cases in a decade, leading to reduced antibiotic use. However, vaccines against many priority drug-resistant pathogens on the WHO list remain unavailable, underscoring the need to expedite research progress in this area [112,113,114]. Researching vaccines against drug-resistant pathogens is hindered by technical challenges, limited market attractiveness, and sluggish development progress. Progress is more noticeable in addressing *Neisseria gonorrhoeae* [115,116]. For example, the protein-based Bexsero^®^ meningococcal vaccine has demonstrated some protective effects against gonorrhea. It is currently undergoing extensive clinical trials for this purpose [117]. As we await the development and assessment of novel vaccines targeting various challenging bacterial pathogens, it remains crucial to encourage the utilization of existing vaccines. Unfortunately, the COVID-19 pandemic has led to immunization program disruptions in over 100 countries, leading to heightened cases of vaccine-preventable diseases and excessive antibiotic usage, especially in LMICs [118,119,120]. To address this concerning trend, it is imperative to enhance healthcare service delivery systems and actively combat vaccine hesitancy.

### 4.5. Healthcare Systems and Sanitation

In areas where health systems are fragile, overburdened, or hard to reach, patients frequently turn to self-care or consult informal healthcare providers due to the restricted availability of formal medical services. Numerous studies conducted in low- and middle-income countries (LMICs) have demonstrated that in situations where healthcare access is limited, antibiotics are often utilized without the supervision of a healthcare expert. For example, in Vietnam, 55.2% of outpatient antibiotics were distributed without a prescription, while the corresponding figure in Bangladesh was 45.7% [121]. Moreover, shortcomings in the infrastructure of the healthcare system create opportunities for other entities to capitalize on the situation, as observed in several countries in South Asia. In countries like India, for instance, pharmaceutical company representatives play a substantial part in increasing sales to wholesalers, who then directly sell to consumers [122]. In areas with weak healthcare systems, patients often self-medicate due to limited access. This creates opportunities for actors like pharmaceutical companies to boost antibiotic sales. In regions where self-medication is common, interventions should target both formal and informal healthcare providers. Addressing these issues requires a strong state apparatus. AMR also stems from social infrastructure gaps in LMICs due to rapid urbanization [123]. The impact of urbanization in low- and middle-income countries has led to the emergence of densely populated urban areas marked by insufficient housing, inadequate access to electricity, limited water treatment, and inadequate waste management. Illustrative instances of this trend include Karachi, which saw its population grow from 5 million to 16 million between 1980 and 2020, Lagos from 2.6 million to 14.4 million, and Manila from 5.95 million to 13.92 million [124,125]. The majority of the population growth in these cities has been concentrated in slum areas, which are characterized by a lack of essential infrastructure to guarantee safe living conditions [124]. AMR implications: Antibiotics foster resistant bacteria, which spread like contagious diseases via contaminated water [126]. Crowded, unhygienic living conditions lead to more infections, prompting higher antibiotic use, both appropriate and inappropriate [127,128]. Willis and Chandler provide an example of a peri-urban slum in Kampala where metronidazole is commonly self-medicated due to rampant diarrheal illnesses caused by inadequate sewage systems. In such dire sanitation situations, the influence of antibiotic use on AMR development might be masked. A study showcased that deteriorating sanitation heightened resistant bacteria risk, regardless of antibiotic exposure levels [129]. In areas with inadequate infrastructure, addressing antibiotic resistance requires investing in both healthcare and sanitation systems. Stewardship efforts should extend beyond education and regulations to include basic infrastructure enhancements that improve overall living conditions. Simply focusing on educating against inappropriate antibiotic use overlooks the systemic issues that drive its necessity.

## 5. Conclusions

To tackle antibiotic overuse in LMICs, we need a comprehensive approach. This involves improving healthcare infrastructure, educating the public about proper antibiotic use, regulating non-prescription sales, training informal healthcare workers, using accessible diagnostics, and promoting vaccines. A global effort is crucial to address these challenges, ensure sustainable changes, and preserve antibiotic effectiveness against infectious diseases.

## 6. Suggestions and Way Forward

Addressing antibiotic overuse in LMICs is complex but crucial. Start by improving healthcare infrastructure, including training and equipping professionals. Rather than only restricting access, address infrastructural weaknesses that drive misuse. Regulations are needed, but must consider disparities in urban and rural areas. Involve informal healthcare workers through training and prescribing permissions. Use affordable POC diagnostic tools integrated into algorithms to guide informed decisions. Stewardship programs require IT investments for effective tracking. Educate healthcare providers and the public, including non-traditional groups, using resources like the WHO’s AWaRe Antibiotic Book. Expand vaccine programs, focusing on accessibility and uptake to reduce infections and antibiotic use. Collaborative efforts are key.

## Figures and Tables

**Figure 1 antibiotics-12-01504-f001:** Factors associated with extensive use.

## Data Availability

Not applicable.

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
