# Peer review of "Empowering Low- and Middle-Income Countries to Combat AMR by Minimal Use of Antibiotics: A Way Forward"

_antibiotics, 2023, doi:10.3390/antibiotics12101504_

Round 1

Reviewer 1 Report

This is an interesting approach but some issues have to be clarified:

Is there enough evidence to proof that the emergence and misuse of antibiotics and spread of AMR is a bigger problem in LMICs compared to HIC? There might be more reference necessary than just reference 3 for this very crucial prerequisite.

Page 5 top: Strategies to improve antibiotic use should involve integrating these workers through recognition, training, and regulation, as demonstrated by an Indian trial [46]. Training didn't significantly alter antibiotic use.

Is the conclusion that training is not fruitful, then?

Generally, is it not an oversimplification to compare all LMIC to HIC? Is it not necessary to stratify for more specific problems, that are typical for some regions but might not be for other regions?  E.g. is lack of regulation and antibiotic stewardship spread all over LMIC?

References, just by curiosity I looked up reference 60 to look for the nano-level detection methods. However, the cited analysis is a meta-analysis about antibiotic stewardship, I could not find data about nano-level detection methods in there. It should be guaranteed that references refer to the statements. (Perhaps I missed it).

Thus, it is advised to double check accuracy of references…

In several cases words are written together that have to be separated.

Author Response

Very grateful to you for your valuable time and struggle to improve the quality of life. all the questions and suggestion are addressed and corrected. Please see the line to line reply  

Is there enough evidence to proof that the emergence and misuse of antibiotics and spread of AMR is a bigger problem in LMICs compared to HIC? There might be more reference necessary than just reference 3 for this very crucial prerequisite.

 More references are incorporated to strengthen the arguments [3-6]

Page 5 top: Strategies to improve antibiotic use should involve integrating these workers through recognition, training, and regulation, as demonstrated by an Indian trial [46]. Training didn't significantly alter antibiotic use.

Is the conclusion that training is not fruitful, then?

Policies and its implementation will work ( Sentence is added)

Generally, is it not an oversimplification to compare all LMIC to HIC? Is it not necessary to stratify for more specific problems, that are typical for some regions but might not be for other regions?  E.g. is lack of regulation and antibiotic stewardship spread all over LMIC?

 In most cases like Pakistan, Africa etc. We'll appreciate , if you suggest some more points or clarity.

References, just by curiosity I looked up reference 60 to look for the nano-level detection methods. However, the cited analysis is a meta-analysis about antibiotic stewardship, I could not find data about nano-level detection methods in there. It should be guaranteed that references refer to the statements. (Perhaps I missed it).

Corrected. Please check Ref 60 and 65

Thus, it is advised to double check accuracy of references…

Regards

Reviewer 2 Report

This is a very well written piece on an important global issue. The authors present the problem/issue well and reference studies from many different countries. While the problem/issue is global in scope, efforts to address it necessarily should be local (one size does not fit all). What may work well in one country, may not in another. What may work well in one part of one country, may not in another part of the same country. It is important to identify the reasons for overprescribing and these may differ between and within countries. What may work in Chile may not in India, for example.

I think that the authors should expand on this theme in their Concluding paragraphs.

It will be up to the editors to determine if this piece presents information already included in other articles in the issue. There is opportunity to shorten this piece so as to avoid repetition in content.

The manuscript highlights the problem and the difficulty in finding solutions (yes, plural, as there probably is not just one solution to the problem). There must be more education for physicians and consumers -- some of the issue is driven by overprescribing by physicians, but another part of the issue is demand by the consumer for antibiotics.

What role should there be for the pharmacist? Perhaps the authors could comment more on this. Also, how "easy" is it to get antibiotics over the counter (eg not from a traditional pharmacy)? Perhaps focusing on this access route could curb the overuse issue.

I enjoyed reading this piece.

Author Response

This is a very well written piece on an important global issue. The authors present the problem/issue well and reference studies from many different countries. While the problem/issue is global in scope, efforts to address it necessarily should be local (one size does not fit all). What may work well in one country, may not in another. What may work well in one part of one country, may not in another part of the same country. It is important to identify the reasons for overprescribing and these may differ between and within countries. What may work in Chile may not in India, for example.

Thanks for your valuable comments

I think that the authors should expand on this theme in their Concluding paragraphs.

Conclusion is modified : To tackle antibiotic overuse in LMICs, we need a comprehensive approach. This involves improving healthcare infrastructure, educating public about proper antibiotic use, regu-lating non-prescription sales, training informal healthcare workers, using accessible di-agnostics, and promoting vaccines. A global effort is crucial to address these challenges, ensure sustainable changes, and preserve antibiotic effectiveness against infectious dis-eases.

It will be up to the editors to determine if this piece presents information already included in other articles in the issue. There is opportunity to shorten this piece so as to avoid repetition in content.

Awaiting editors comments on it , if any

The manuscript highlights the problem and the difficulty in finding solutions (yes, plural, as there probably is not just one solution to the problem). There must be more education for physicians and consumers -- some of the issue is driven by overprescribing by physicians, but another part of the issue is demand by the consumer for antibiotics.

Training of health care providers is suggested

What role should there be for the pharmacist? Perhaps the authors could comment more on this. Also, how "easy" is it to get antibiotics over the counter (eg not from a traditional pharmacy)? Perhaps focusing on this access route could curb the overuse issue.

Sale without prescription is common phenomenon without the distinction in the pharmacy nature in LMICs

I enjoyed reading this piece

Regards